# Antioxidant Activity, Stability in Aqueous Medium and Molecular Docking/Dynamics Study of 6-Amino- and *N*-Methyl-6-amino-L-ascorbic Acid

**DOI:** 10.3390/ijms24021410

**Published:** 2023-01-11

**Authors:** Lara Saftić Martinović, Nada Birkic, Vedran Miletić, Roberto Antolović, Danijela Štanfel, Karlo Wittine

**Affiliations:** 1Department of Biotechnology, University of Rijeka, Radmile Matejčić 2, 51000 Rijeka, Croatia; 2Group for Applications and Services on Exascale Research Infrastructure, Faculty of Informatics and Digital Technologies, University of Rijeka, Radmile Matejčić 2, 51000 Rijeka, Croatia; 3Department of Nursing, School of Medicine, Catholic University of Croatia, Ilica 242, 10000 Zagreb, Croatia; 4Jadran-Galenski Laboratorij d.d., Svilno 20, 51000 Rijeka, Croatia; 5Selvita Ltd., Prilaz Baruna Filipovića 29, 10000 Zagreb, Croatia

**Keywords:** ascorbic acid, antioxidant, stability, peroxiredoxin, molecular docking/dynamics, ABTS, DPPH, MS/MS

## Abstract

The antioxidant activity and chemical stability of 6-amino-6-deoxy-L-ascorbic acid (D1) and *N*-methyl-6-amino-6-deoxy-L-ascorbic acid (D2) were examined with ABTS and DPPH assays and compared with the reference L-ascorbic acid (AA). In addition, the optimal storing conditions, as well as the pH at which the amino derivatives maintain stability, were determined using mass spectrometry. Comparable antioxidant activities were observed for NH-bioisosteres and AA. Moreover, D1 showed higher stability in an acidic medium than the parent AA. In addition, AA, D1, and D2 share the same docking profile, with wild-type human peroxiredoxin as a model system. Their docking scores are similar to those of dithiothreitol (DTT). This suggests a similar binding affinity to the human peroxiredoxin binding site.

## 1. Introduction

L-ascorbic acid (vitamin C, AA, 2,3-dehydro-L-gulonic acid-γ-lactone) is a small hydrophilic molecule which is soluble in water. It takes part in many biochemical processes in organisms and is one of the most effective natural antioxidants; it is a key player in the scavenging of reactive oxygen-based (or nitrogen-based) radical species [1,2,3]. This property is related to at least three factors: (1) the redox potential of the environment; (2) the presence/absence of transition metals; and (3) the local (cellular or plasma) concentrations of ascorbate [4,5,6,7]. The electron-rich lactone 2,3-endiol system enables AA to donate one or two electrons. Under physiological conditions AA exists as the resonance-stabilized ascorbate anion formed by deprotonation of the hydroxyl group at C3 [8,9]. Exposure to air, light, heat, water, transition metal ions such as Cu^2+^ and Fe^2+^, and alkaline pH conditions causes the reversible oxidation of ascorbic acid to dehydroascorbic acid. Dehydroascorbate can be hydrolyzed to 2,3-diketo-L-gulonate, which is spontaneously degraded to oxalate, CO_2_, and L-erythrulose [1,10,11]. On the other hand, oxidation leads to a range of products, such as l-threonic acid (ThrO), oxalic acid (OxA), and their esters [12,13,14]. Ascorbic acid retains maximum stability near pH 3.0 and pH 6.0 [15]. Alternatively, under anaerobic conditions in aqueous solutions ascorbic acid is dehydrated and hydrolyzed to give furfural and carbon dioxide. The rate of degradation is maximum at pH 4.1, corresponding to the pKa of AA [16].

One way to improve the stability of AA is to chemically modify the structure. The introduction of a phosphate group in the form of sodium (SAP) or magnesium (MAP) salts at the O2 position increases the oxidative stability [17]. On the other hand, glucose at the O2 position confers protection against degradation at high temperatures and pH and the action of metal ions [18]. Chemical stability is increased in the lipophilic derivatives of AA, such as ascorbyl 6-palmitate (AA-Pal), and tetra-isopalmitoyl ascorbic acid (IPAA) (Figure 1) [19]. Known modifications generally a cause loss of biological activity, and in vivo enzymatic conversion to AA is necessary for re-activation.

Although AA has poor molecular stability [20,21,22], recent density functional theory (DFT) calculations confirmed a better antioxidant performance in the aqueous phase compared to the derivatives with increased stability, such as ascorbyl 2-glucoside (AA2G), 3-*o*-ethyl-l-ascorbic acid (AAE), and ascorbyl 6-palmitate (AA6P) (Figure 1). The calculated results suggest that the solvent effect influences the antioxidant capacity of AA and its derivatives [23].

Generally, the antioxidant properties of AA as a scavenger of free radicals are related to its ability to form a stabilized radical. Greater antioxidant stability could have a positive effect on a range of the cellular process in which AA takes part. AA exhibits its antioxidant and anti-inflammatory properties at various levels of the functioning of living cells, starting from the direct scavenging of free radicals (hydroxyl and superoxide anion) [24], the silencing of pro-inflammatory pathways (e.g., NF-kB) [25], and blocking the transcription of cytokines (IL-2, IL-6, IL-12, and TNF-α). It also activates intracellular antioxidant systems (superoxide dismutase, SOD; glutathione peroxidase, GSH-Px; glutathione reductase, GSSG-R, TRxR; and catalase, CAT) [26,27,28,29,30,31]. The activation of GSH-Px and GSSG-R is a ground reason for the hepatoprotective and gastroprotective effects of ascorbate [32,33]. Furthermore, the induction of TrxR activity by ascorbate is accompanied by the stimulation of the GSH-based antioxidant system [34]. The effect of ascorbic acid on TrxR activity may be an encouraging tool in cancer therapies focused on TrxR hyperactivity [35]. In rapidly proliferating mammalian cells, such as gastric cancer cells, glioblastoma, and carcinoma cells, it is observed that AA reduces CAT activity and causes an increase in the level of hydrogen peroxide, allowing the molecule to perform the signaling function [28,36]. The main pro-inflammatory signaling pathway affected by AA is the NFκB/TNFα pathway [25,37,38,39,40,41]. AA downregulates the expression of pro-inflammatory factors (IL-6, IL-12, and TNF-α) and upregulates anti-inflammatory cytokines (IL-4 and IL-10) in mouse splenocytes [42]. Additionally, AA (2.3 mM) reduces IL-2 and IL-6 production through the reduction in the proliferation of mononuclear cells in porcine peripheral blood [43,44]. AA (0.25–1 mM) induces apoptosis in melanoma or acute myeloid leukemia cells through Bcl-2 overexpression and caspase 3 and 9 activation [45,46]. AA also interacts with cellular antioxidants (glutathione, thioredoxin, coenzyme Q, α-tocopherol, and retinol) [47,48,49,50] and supports the action of other exogenous antioxidants (rutin, phenolic acid, curcumin, green tea polyphenol epigallocatechin-3-gallate, gallic acid, and xanthone) [51,52,53].

It is thus of considerable importance to find derivatives that possess antioxidative activity comparable to or higher than the parent AA and/or better oxidation resistance, which could give them a practical value. In this context, we studied the influence of the bioisosteric replacement of OH with NH_2_ at C6 of AA, mainly 6-amino-6-deoxy-L-ascorbic acid (NH_2_AA, D1) and *N*-methyl-6-amino-6-deoxy-L-ascorbic acid (CH_3_NHAA, D2). A wild-type human peroxiredoxin (PDB ID: 3MNG) which helps in scavenging peroxide and is involved in the metabolic cellular response to reactive oxygen species was chosen as a model system for the docking and molecular dynamics studies [54,55].

## 2. Results

The L-ascorbic acid derivatives D1 and D2 were synthesized according to Šušković et al. [56,57]. The structures were confirmed by ^1^H NMR (Appendix A).

### 2.1. Stability Measured by Spectrophotometric Methods

The antioxidant activity and antioxidant stability of D1 and D2 were analyzed. In addition, the stability was tested in different pH environments for 30 days. Initially, the pH values of the sample solutions were measured to set the pH values of the solutions which were intended for the stability analysis. The pH of the AA dissolved in Milli-Q water was 3.1, while the pH values of the D1 and D2 derivatives were 5.9 and 4.2. Therefore, for the purpose of the stability analysis, the pH value of 2.8 was chosen for the acidic medium, while the pH value of 12 was chosen for the alkaline medium (and 7 was the pH value of the neutral medium).

In order to evaluate the stability of D1 and D2, we analyzed two very stable derivatives of AA that are often used in the cosmetic industry: sodium ascorbyl phosphate (SAP) and magnesium ascorbyl phosphate (MAP) (Appendix A). The results of the ABTS antioxidant activity showed that D1 and D2 have antioxidant activity similar to that of the parent AA, whose IC_50_ value was 3.89 µg/mL (Figure 2). In addition, D1 with an IC_50_ value of 4.45 µg/mL and D2 with an IC_50_ value of 5.65 µg/mL showed higher antioxidant activity than the most commonly used SAP and MAP derivatives, with IC_50_ values of 28.78 and 38.52 µg/mL, respectively (Figure 2 and Appendix A).

To confirm the results of the antioxidant activity obtained by the ABTS method, the DPPH method was employed. The DPPH analysis confirmed that D1 is a more potent antioxidant than D2. The obtained IC_50_ value of D1 is 3.87 µg/mL, while for D2 it is 7.51 µg/mL (Figure 3).

In order to test the antioxidant stability of D1 and D2 over time in a neutral medium, the ABTS method was performed after 0, 1, 2, 3, 7, 15, 21, and 30 days of incubation at 20 °C and at +4 °C in the dark (Figure 4 and Figure 5). The antioxidant stability of the 6-amino-L-ascorbic acid derivatives was compared to the antioxidant stability of AA. AA was degraded at 20 °C in the dark after the 2nd day of incubation. The results show that the newly synthesized derivatives retained their antioxidant stability until the 3rd day of incubation, after which they also degraded. The incubation of the samples at +4 °C in the dark prolonged the stability of AA, D1, and D2. Stability until the 15th day of incubation was maintained for D2 (Figure 5). For comparison, the SAP and MAP derivatives showed stability for 30 days in both tested conditions (the MAP derivative started to degrade after 30 days of incubation at 20 °C in the dark) (Appendix A).

Stability was also assessed with the DPPH method. The results showed that at 20 °C in the dark, D2 was stable until the 2nd day of incubation (Figure 6). D1 showed stability until the 3rd day of incubation. The results of the DPPH method after storing the samples at +4 °C in the dark showed that D1 was extremely antioxidant-stable during the 30 days of the experiment (Figure 7). D2 showed antioxidant stability up to the 7th day of incubation.

### 2.2. Stability Measured by MS/MS

The stability of D1 and D2 was tested by mass spectrometry in neutral, acidic, and alkaline medium at +20 °C and +4 °C in the dark. The highest stability of D1 and D2 was detected in an acidic environment at +4 °C in the dark. The initial value of the area under the curve (AUC) of each compound was taken as the maximum concentration of the compound in the sample (100%). During the 30-day period, the AUC value decreased (degradation of the analyte), and its values were converted into the percentage of analyte remaining in the sample. The stability in the alkaline solvent was negligible considering that it was not possible to detect the analyte on day 0 of the analysis. The stability tested by MS proved a rapid degradation of AA in the neutral medium at +20 °C and +4 °C in the dark. In an acidic medium, especially at +4 °C in the dark, AA retained its stability. Under these conditions, after 30 days of incubation, 26% of the analyte in the sample was detected. At +20 °C in the dark, AA degraded after day 0 of the analysis in a neutral solvent, that is, after the 7th day of the analysis in an acidic solvent (Figure 8). In an acidic solvent at +4 °C in the dark, the stability of D1 and D2 was maintained. After the 30th day of incubation, 85% of D1 and 37% of D2 were detected in the samples (Figure 8). At 20 °C in the dark in a neutral solvent, D1 was completely degraded after the 3rd day, and D2 after the 2nd day of analysis. At the same temperature in an acidic solvent, D1 was detectable during all 30 days of analysis, and D2 was degraded after the 3rd day of analysis.

The precursor ion of D1 is 175.8 *m*/*z* in the positive recording mode (Appendix A). The ion intensity after 30 days of experiment indicated approximately the same concentration level of D1 as on day 0 of the experiment. This is in correlation with the previous results that showed 85% of the detected analyte in the sample after 30 days of analysis (Figure 8). The precursor ion of D2 was 189.9 *m*/*z* in the positive imaging mode. After the 30th day of analysis, the intensity of the 189.9 ion visibly decreased, while the intensity of the decay product ions increased (Appendix A). In contrast to the marked stability of D1, D2 mostly decomposed, and after 30 days of analysis, 37% of the derivative remained in the sample.

### 2.3. Docking Control Study of the DTT against Wild-Type Human Peroxiredoxin

We re-evaluated crystal DTT as a potential wild-type human peroxiredoxin inhibitor and performed an extensive free docking run on it. According to the RxDock documentation [58], the binding modes of the ligands are ranked according to their docking score based on the master docking function, which is a weighted sum of the intermolecular, ligand intramolecular, site intramolecular, and external restraint terms. Table 1 lists the docking score and structural data for DTT. Unlike our previous work [59], where we used the weighted sum of the intermolecular score, in this paper we used the total RxDock docking score, including all the components of the intramolecular and intermolecular scoring functions. For DTT, the score for the most favorable pose was −26.335.

### 2.4. Molecular Docking Study of the Compound Library

At the end of the protocol described in the Materials and Methods section, the compounds presented in Table 1 were selected based on the total RxDock score.

AA, D1, and D2 share the same docking profile. Their docking scores are similar and within the range of −26.641 and −25.576. They share a similar docking score to DTT with 3MNG of −26.335 (Table 1). This suggests a similar binding affinity to the human peroxiredoxin binding site.

The hydrogen bonds that these compounds make with the human peroxiredoxin binding site also share a similar profile. All the compounds of interest form between six and seven hydrogen bonds with Thr44, Gly46, Cys47, and Thr147 (Figure 9 and Figure 10).

### 2.5. Molecular Dynamics Study of the Compound Library

The molecular dynamics study of the compound library suggests non-specific binding to wild-type human peroxiredoxin. While all of the compounds remain firmly bound to the protein itself, they do change multiple conformations within the 10 ns period. Figure 11 shows the RMSD values of all the ligands of interest, which share an interactive profile in which the RMSD stays below 0.15 RMSD across the board.

This further suggests the equivalency of the binding modes of AA with D1 and D2.

## 3. Discussion

This study evaluated the antioxidant activity and stability of D1 and D2 compared to AA. In addition, the favorable storage conditions of the D1 and D2 solutions and the pH value at which the stability was maintained were determined. It should be noted that the stability of the DPPH and ABTS radicals is also pH-dependent [61]. Thus, the results of the antioxidant activity and stability obtained by these methods served as preliminary findings, and the stability was additionally checked by the MS/MS method. The stability in an acidic pH was examined to eliminate the performance issues of ABTS and/or DPPH at a low pH.

The spectrophotometric and subsequent MS/MS analysis showed that D1 and D2 had a somewhat weaker antioxidant activity compared to AA but a higher stability in an acidic medium. D1 showed a significant improvement in stability over a period of 30 days (85% of the sample was otherwise stable). In comparison, the stability of the D2 derivative after 30 days in an acidic medium was 37%. D1 also proved to be more stable in a neutral medium. This could be attributed to the hydrogen bonding network stabilization (D1 contains an NH_2_ group while D2 contains a CH_3_NH group (6 H_b_ donors for D1 vs. 5 H_b_ donors for D2)).

The IC_50_ values obtained by the ABTS method show that AA is the most potent antioxidant (3.8 µg/mL). Although D1 and D2 showed a somewhat lower antioxidant activity (4.4 µg/mL for D1 and 5.6 µg/mL for D2), their antioxidant activity is comparable to the antioxidant activity of AA (IC_50_ is significantly lower in comparison to SAP and MAP). The high IC_50_ values of SAP and MAP are the result of their stability. It was previously shown that the antioxidant activity of the AA derivatives is inversely proportional to the degree of their stability [23]. Corresponding IC_50_ values of 3.57 µg/mL for AA, 3.87 µg/mL for D1, and 7.51 µg/mL for D2 were obtained using the DPPH method.

The DPPH antioxidant activity and the ABTS antioxidant activity for the same derivative can differ from each other (and finally from the results obtained by MS/MS analysis) due to the instability of the radicals that deteriorate in light. Therefore, the results depend on the speed of the analysis. The stability could also be influenced by the electron-donating methyl group (e.g., the ABTS method for D2). The concentration of substrate and the antioxidant mechanism could also be a contributing factor to the stability.

AA and D1 have the same number of polar contacts, while AA has a noticeably more favorable inter-polar score. Interestingly, D2 has one polar contact more than AA and D1, and the inter-polar docking score lies between those two ligands. Overall, is concluded from the docking studies that all three of the ligands bind favorably to the binding site. However, the total score, in this case, is just an initial preliminary result since we are primarily interested in the hydrogen bonds between the ligand and the receptor. As the molecules are too similar, it is not possible to make a strong conclusion on the docking study. For this reason, the stability of the AA complexes and derivatives with 3MNG were analyzed. The results showed similar binding modes of AA and the newly synthesized derivatives, suggesting that the newly synthesized derivatives have not lost their biological activity compared to AA.

### Possible Applications

AA is well known for its ability to stimulate the synthesis of collagen [62] and suppress the pigmentation and decomposition of melanin [17]. Currently, two of the most frequently used derivatives in cosmetics are SAP and MAP, which have weaker antioxidant activity, but maintain long-term stability. These derivatives are stable in emulsions with a pH between 6 and 7. Even though they are frequent ingredients of cosmetic products, they are poorly absorbed through the skin [63].

The effectiveness of a cosmetic formulation depends on its pH. The pH value of healthy skin is in the range of 4.0 to 6.0 [64]. For the formulation to be suitable for an industrial production, its pH value should be within the specified range. Therefore, the development of new AA derivatives with activity similar to or stronger than AA is desirable, while ensuring the long-term stability of the product and lowering the pH values in which they remain stable. In addition, most of the previously described derivatives require in vivo conversion to AA to exhibit the desired effect [63].

The pKa1 value of derivative D1 is 3.7, and the pKa1 value of derivative D2 is 3.9 (calculations made with Percepta 2019.1.0 program (ACD Labs)), and at pH values lower than their pKa1, the molecules are positively charged. Positively charged molecules in topical formulations may lead to improved penetration due to increased interaction with the negatively charged membrane [65,66]. However, extended research on the NH-bioisosteres of AA is necessary in order to further evaluate their cosmetic potential, especially regarding stability, in vivo action, and the form or carriers that could enhance penetration into the skin layers.

## 4. Materials and Methods

### 4.1. Reagents

(±)-6-hydroxy-2,5,7,8-tetramethyl-chroman-2-carboxylic acid (TROLOX) was purchased from Sigma Aldrich (St. Louis, MO, USA); 2,2′-azino-bis (3-ethylbenzothiazoline-6-sulfonic acid (ABTS) and 2,2-diphenyl-1-picrylhydrazyl (DPPH) from Alfa Aesar (Thermo Fisher Scientific, Waltham, MA, USA); potassium persulfate from VWR Chemicals BDH (Radnor, PA, USA); and L-(+)-ascorbic acid from Scharlab S.L. (Barcelona, Spain).

Ethanol, methanol, and acetonitrile, all LC-MS grade, were purchased from Sigma Aldrich. Formic acid (LC-MS grade) was purchased from Honeywell Research Chemicals (Muskegon, MI, USA).

Milli-Q water was obtained using Ultrapure Water Systems (GenPure UV-TOC/UF × CAD plus) connected to the Milli-Q Water Purification System (<0.055 μS/cm, Milli-Q Model Pacific TII 12, Thermo Scientific, Waltham, MA, USA, NOW).

### 4.2. NMR

The NMR spectra of the L-ascorbic acid derivatives were recorded using a Bruker AV600 high-resolution instrument (B0 = 14.1 T) at a temperature of 25 °C. The samples were dissolved in deuterated dimethylsulfoxide (DMSO-*d*_6_) with 0.03% tetramethylsilane or deuterated water (D_2_O) with 0.03% tetramethylsilane. The spectra were recorded at a frequency of 600.130 MHz and 150.903 MHz for ^1^H-atom nuclei. In relation to the chemical shift of the tetramethylsilane (TMS) signal, the chemical shifts (δ/ppm) of the signals in the spectra of the samples dissolved in DMSO-*d*_6_ or D_2_O (δTMS = 0.0 ppm) were determined.

### 4.3. Sample Preparation for Measurements of Antioxidant Stability

AA, D1, and D2 were dissolved in Milli-Q water at a concentration of 1 mM. The Trolox stock solution was diluted in ethanol at concentrations of 0.03, 0.06, 0.09, 0.12, 0.15, 0.18, and 0.21 mM. The stock solution of AA was diluted in water in concentrations of 0.03, 0.06, 0.09, 0.12, 0.15, 0.18, 0.21, 0.24, 0.27, 0.30, 0.5, 0.7, and 1 mM. The stock solutions of D1 and D2 were diluted in water at concentrations of 0.03, 0.06, 0.09, 0.12, 0.15, 0.18, 0.21, 0.24, 0.20, and 0.30 mM.

A set of dilutions of each sample was prepared twice. One set of samples during the experiment was stored at room temperature (+20 °C) in the dark, and the other set of samples was stored at +4 °C in the dark. To analyze the stability of the samples, they were analyzed after 0, 1, 2, 3, 7, 15, 21, and 30 days.

#### 4.3.1. ABTS

For the ABTS assay, a 7 mM solution of ABTS and a 2.4 mM solution of potassium persulfate in water were mixed at a ratio of 1:1 (*v*/*v*) and incubated for 14 h in the dark at room temperature. Before each measurement, the prepared ABTS radical solution was diluted in methanol to an absorbance of 0.7. Forty microliters of each sample in dilutions was mixed with 160 µL of ABTS solution, and absorbance readings were performed at a 734 nm wavelength after 7 min of incubation using a TECAN Monochromator Infinite M200 Pro. Measurements were performed after 0, 1, 2, 3, 7, 15, 21, and 30 days of storage. An example of the ABTS test is shown in Appendix A.

The sample concentration required to reduce the radical absorbance by 50% (IC_50_) was calculated from the absorbance data after 7 min of incubation. The data are expressed as the mean ± standard deviation (SD).

#### 4.3.2. DPPH

For the DPPH assay, a 0.1 mM solution of DPPH in ethanol was prepared. Forty microliters of each sample in dilutions and 160 µL of the DPPH solution were mixed and absorbance was measured at a 517 nm wavelength after 30 min of incubation using a TECAN Monochromator Infinite M200 Pro. Measurements were performed after 0, 1, 2, 3, 7, 15, 21, and 30 days. An example of the DPPH test is shown in Appendix A.

The sample concentration required to reduce the radical absorbance by 50% (IC_50_) was calculated from the absorbance data after 30 min of incubation of the samples in the dark. The data are expressed as the mean ± standard deviation (SD).

### 4.4. MS/MS Stability Analysis

For MS/MS analysis by direct injection of AA, D1, and D2, the Agilent 1260 series HPLC chromatograph (equipped with a degasser, a binary pump, and an autosampler) and a column oven coupled to the Agilent 6460 triple quadrupole mass spectrometer (equipped with Jet Stream electrospray source) was used (Agilent Technologies, Palo Alto, CA, USA).

The analysis was performed by scanning the precursor and product ions. Based on the obtained results, the multiple reaction monitoring (MRM) method was developed. The parameters of the developed MRM method are shown in Table 2. In addition to the MRM method, an MS scan method was developed with the same parameters of the MS system.

The mobile phase consisted of (A) 0.1% formic acid in milli-Q and (B) 0.1% formic acid in ACN. The isocratic flow of 50% mobile phase A and 50% mobile phase B was used for the analysis. The flow rate was 0.2 mL/min. For each analysis, 1 μL of the prepared sample was injected. For AJS-ESI-QQQ, the parameters were set as follows: capillary voltage 3.5 kV in positive and negative modes, nozzle voltage 0.5 kV, ion source temperature 300 °C, gas flow 5 L/min, pressure 45 psi, drying gas temperature 250 °C, and gas flow 11 L/min. Nitrogen was used as the gas in the collision cell. Collision energies were adjusted from 6 V to 18 V depending on the compound analyzed.

The samples were prepared in a neutral, acidic (pH 2.8, adjusted using formic acid) and alkaline medium (pH 12, adjusted using sodium hydroxide) in a concentration of 5 ppm. Each sample was injected in triplicate, and the analyses were performed 0, 1, 2, 3, 7, 15, 21, and 30 days after sample preparation. The sample stability was estimated from the intensity of the quantifier ion.

### 4.5. Molecular Docking

Molecular docking methods are used to predict the interactions of drugs with macromolecules. The blind docking method involves a search throughout the entire surface of a macromolecule for binding sites. Therefore, a blind molecular docking analysis of a customized small ligand library of compounds of interest was performed.

#### 4.5.1. Protein Preparation

The crystal structure of wild-type human peroxiredoxin with DTT bound as a competitive inhibitor (PDB ID: 3MNG) was obtained from the Protein Data Bank (PDB) [67]. Crystal water was removed from the structure, as well as all other solvent artifacts, leaving just the protein residues.

This protein was minimized using the GROMACS software suite. The protein was parameterized using the AMBER99SB-ILDN protein, nucleic AMBER94 force field [68]. For water topology, the TIP3P three-site model was used. A dodecahedron system cell was defined, with 1.5 Angstrom between the solute and the generated cell. After the system was defined, the protein was solvated and neutralized. The system was then minimized using the steepest descent minimization protocol.

The protein prepared in this manner was used in the subsequent docking simulations.

#### 4.5.2. Cavity Search

The RxDock software package [59] was used to search and define the binding site of minimized human peroxiredoxin. The reference ligand method was used, with a crystal DTT bound as a competitive inhibitor serving as a reference.

#### 4.5.3. (4S,5S)-1,2-Dithiane-4,5-diol (DTT) Optimization and Docking

DTT was re-docked and scored to confirm that the docking software could adequately predict the binding conformations of the subsequent compound library in a given binding site.

DTT was thus docked to the wild-type human peroxiredoxin binding site using a full docking protocol that consisted of three stages of genetic algorithm search, followed by low-temperature Monte Carlo and simplex minimization stages per each generated pose. The exhaustiveness of a hundred poses for DTT was used and then ranked according to the measured binding affinity (score). The most favorable one was used as a template for the subsequent HTVS analysis.

#### 4.5.4. Customized Small Ligand Library Preparation and High-Throughput Virtual Screening

Compounds of interest, namely AA, NH_2_AA (D1), and CH_3_NHAA (D2), were optimized for docking. This library was docked to a wild-type human peroxiredoxin binding site using the same protocol as in case of the DTT docking (Section 4.5.3).

### 4.6. Molecular Dynamics Simulation Studies

The molecular dynamics (MD) simulation studies between the selected compounds and the wild-type human peroxiredoxin were performed for the period of 10 ns using the GROMACS software suite [69], version 2022.1. The protein–ligand complexes were acquired from the high-throughput virtual screening phase of the research and were considered to be the starting point of the MD simulations. The AMBER99SB-ILDN protein, nucleic AMBER94 force field [68] was used to establish the complex stability. The ligand-receptor parameters were calculated using GROMACS, and the ligands custom parameters were calculated using AnteChamber PYthon Parser interfacE (ACPYPE) [70].

The ligand–protein complexes were prepared and solvated using the tip3p/SPC216 water model [71] in a dodecahedral cell. Na^+^ or Cl^−^ ions were used to neutralize the system. All the complexes were primarily energy-minimized with the steepest descent method [72], followed by two sequential equilibration simulations using a canonical (NVT) and isobaric–isothermal (NPT) ensemble for 100 picoseconds (ps) each. The temperature for the Maxwell distribution was set to 300 Kelvins, and the Berendsen thermostat was set to 1 bar of pressure. Using the NPT group, a final MD simulation was carried out and long-range electrostatic interactions were identified by the particle mesh Ewald (PME) method [73]. The simulation was analyzed using internal GROMACS programs and scripts. The total interaction energy was calculated as a sum of the short-range Coulomb interaction and the short-range Lennard-Jones interaction between the protein residues and the ligands.

### 4.7. Data Processing and Statistical Analysis

The statistical data processing was performed in MassHunter Qualitative analysis version B.07.00 (Agilent Technologies, Santa Clara, CA, USA). The analysis and the final presentation of the results was performed in Excel (version 2012, Microsoft, Redmond, WA, USA). The Percepta 2019.1.0 program (ACD Labs, Toronto, Ontario, Canada) was utilized to predict the protonated/deprotonated state of AA, D1, and D2 at different pH values.

## 5. Conclusions

In this paper, the antioxidant activity and stability of 6-amino-6-deoxy-L-ascorbic acid (D1) and *N*-methyl-6-amino-6-deoxy-L-ascorbic acid (D2) were examined with ABTS and DPPH chemical assays. In addition, the optimal storage conditions were examined, and the pH values at which the derivatives maintained stability were determined by MS/MS analysis. D1 demonstrated increased antioxidant stability if stored at +4 °C in the dark. The MS/MS analysis confirmed that after 30 days 85% of D1 was still present in the analyzed sample. Furthermore, L-ascorbic acid, D1, and D2 have the same docking profile as wild-type human peroxiredoxin. Their docking scores are similar to those of dithiothreitol (DTT). This suggests a similar binding affinity to the human peroxiredoxin binding site. This makes them good candidates for further in vitro and in vivo evaluation of the antioxidant and anti-inflammatory properties (see Introduction Section).

## Figures and Tables

**Figure 1 ijms-24-01410-f001:**
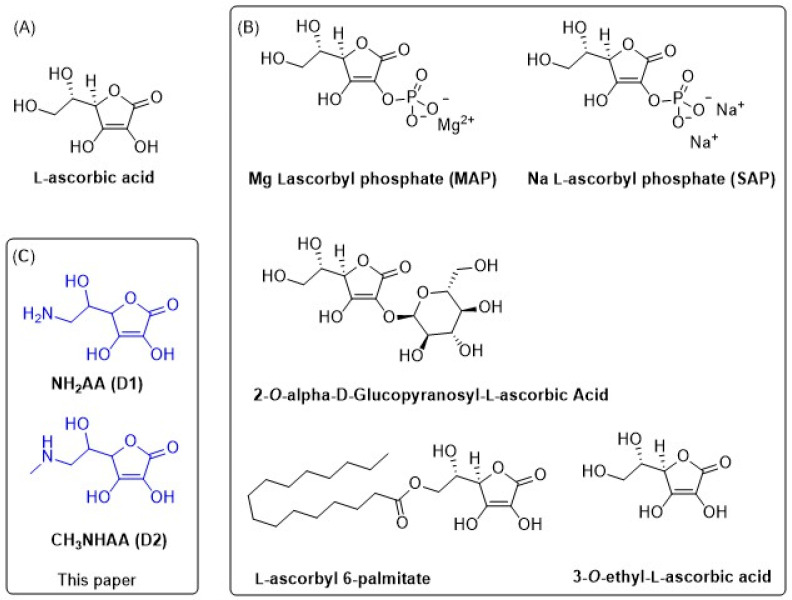
Structures of L-ascorbic acid (**A**), widely known and previously studied derivatives of L-ascorbic acid (**B**) and herein studied 6-amino-6-deoxy-L-ascorbic acid (NH_2_AA, (**C**)) and *N*-methyl-6-amino-6-deoxy-L-ascorbic acid (CH_3_NHAA, (**C**)).

**Figure 2 ijms-24-01410-f002:**
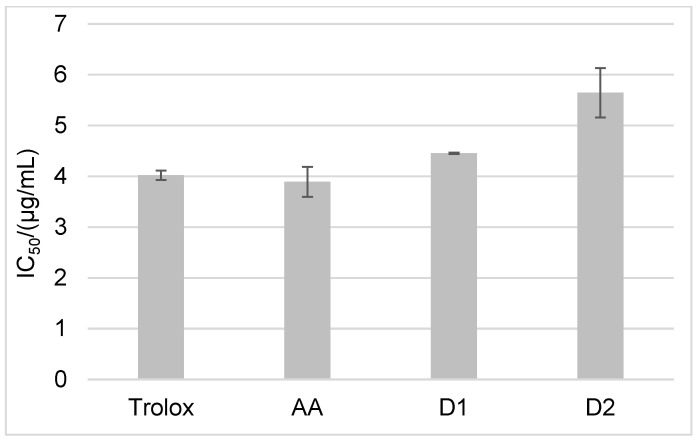
ABTS antioxidant activity of Trolox, AA, NH_2_AA (D1), and CH_3_NHAA (D2) on day 0. Results are expressed as µg/mL ± standard deviation (SD).

**Figure 3 ijms-24-01410-f003:**
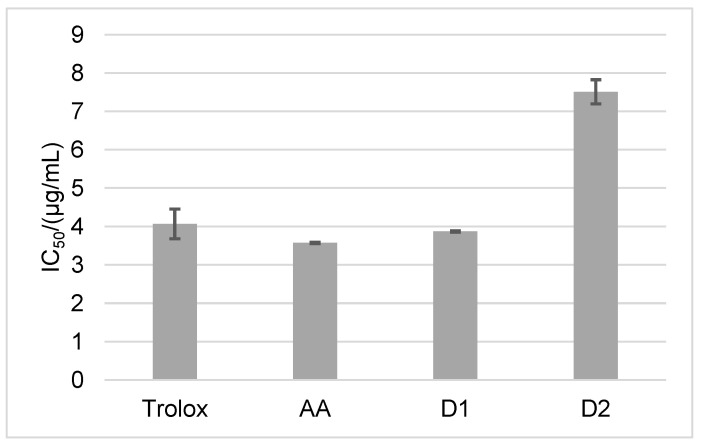
DPPH antioxidant activity of Trolox, AA, NH_2_AA (D1), and CH_3_NHAA (D2) on day 0. Results are expressed as µg/mL ± standard deviation (SD).

**Figure 4 ijms-24-01410-f004:**
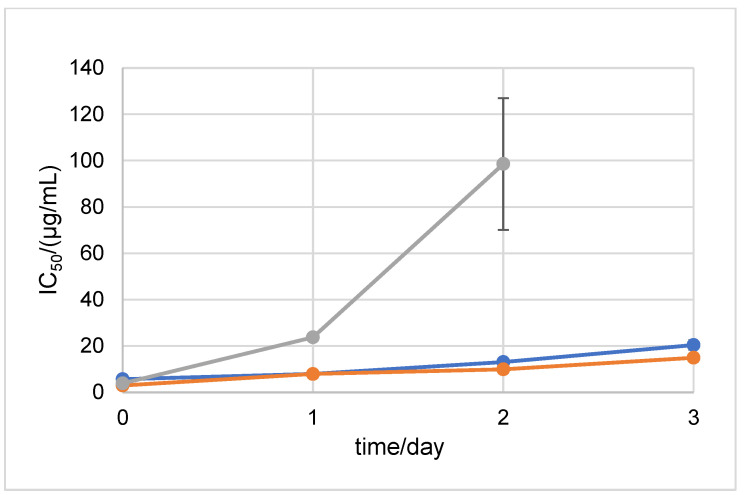
ABTS antioxidant stability of AA, NH_2_AA (D1), and CH_3_NHAA (D2) after 0, 1, 2, and 3 days of incubation at 20 °C in the dark (after 3 days, all constituents decomposed). Results are expressed as µg/mL ± standard deviation (SD). D1 is shown in orange, D2 in blue, and AA in gray.

**Figure 5 ijms-24-01410-f005:**
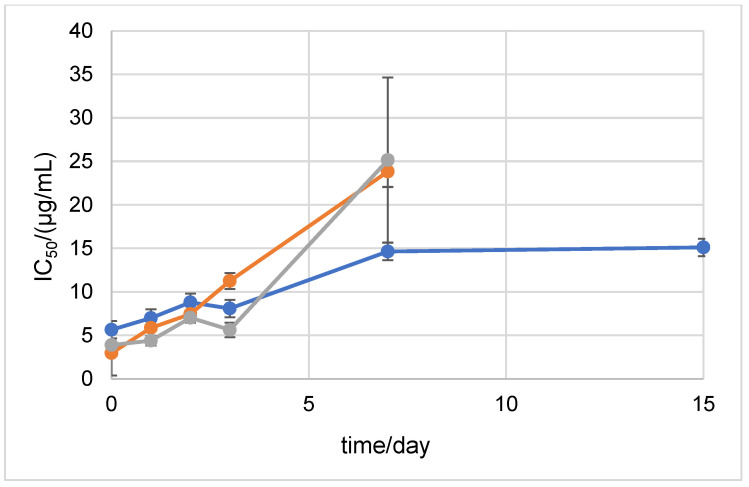
ABTS antioxidant stability of AA, NH_2_AA (D1), and CH_3_NHAA (D2) after 0, 1, 2, 3, 7, and 15 days of incubation at +4 °C in the dark (after 15 days, all constituents decomposed). Results are expressed as µg/mL ± standard deviation (SD). D1 is shown in orange, D2 in blue, and AA in gray.

**Figure 6 ijms-24-01410-f006:**
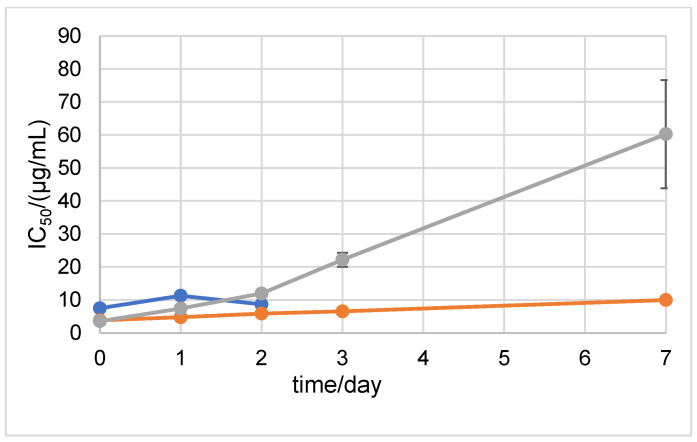
DPPH antioxidant stability of AA, NH_2_AA (D1), and CH_3_NHAA (D2) after 0, 1, 2, 3, and 7 days of incubation at 20 °C in the dark (after 7 days, all constituents decomposed). Results are expressed as µg/mL ± standard deviation (SD). D1 is shown in orange, D2 in blue, and AA in gray.

**Figure 7 ijms-24-01410-f007:**
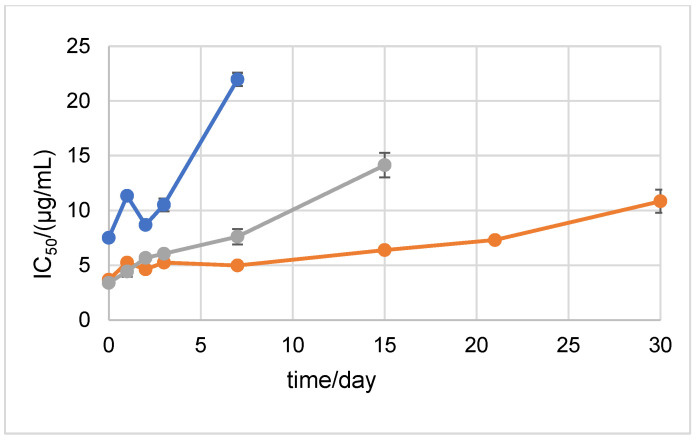
DPPH antioxidant stability of AA, NH_2_AA (D1), and CH_3_NHAA (D2) after 0, 1, 2, 3, 7, 15, 21, and 30 days of incubation at +4 °C in the dark. Results are expressed as µg/mL ± standard deviation (SD). D1 derivative is shown in orange, D2 in blue, and AA in gray.

**Figure 8 ijms-24-01410-f008:**
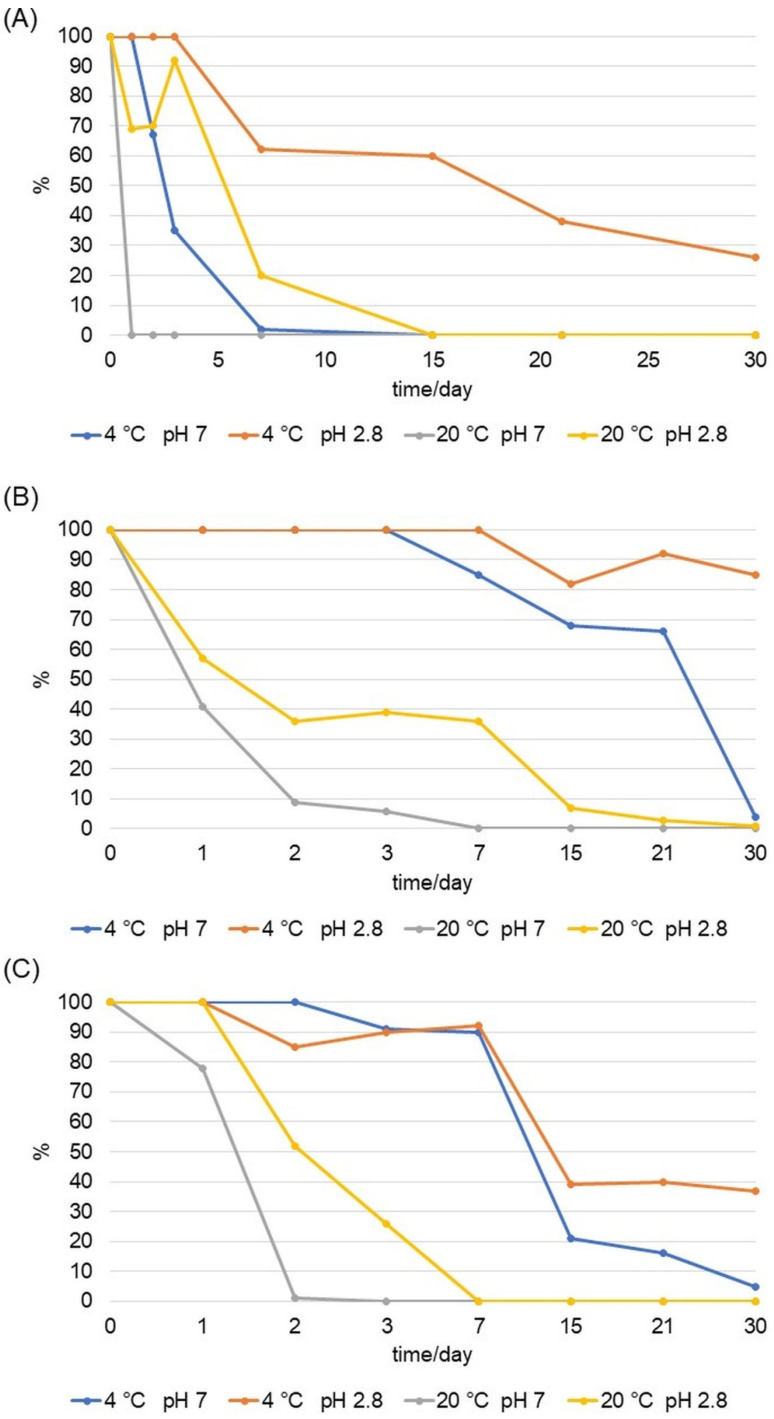
MS/MS stability of AA (**A**), NH_2_AA (D1) (**B**), and CH_3_NHAA (D2) (**C**) in neutral, acidic, and alkaline solvent. The figure shows the temperature (T), the pH of the compound solution, the percentage of the analyte detected in the sample, and the day of the analysis.

**Figure 9 ijms-24-01410-f009:**
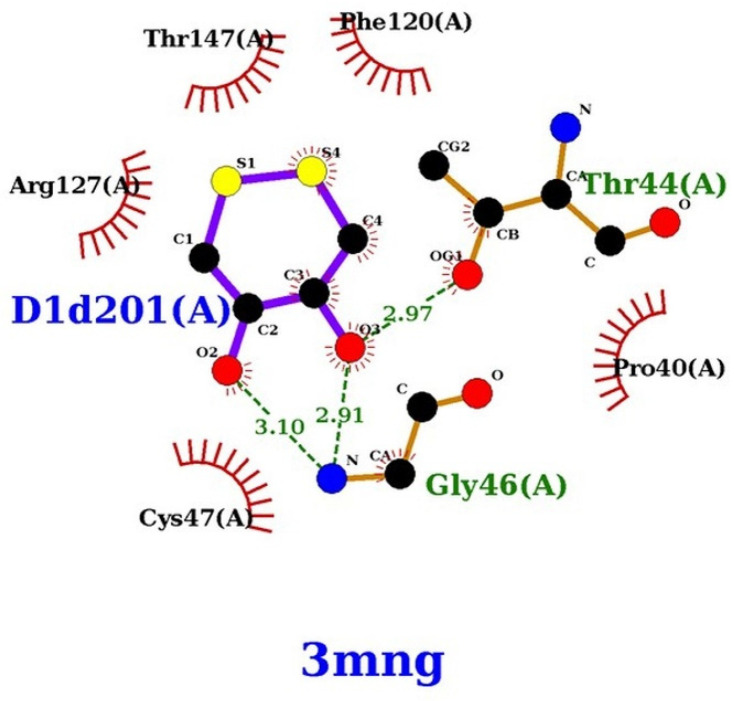
DTT interaction with wild-type human peroxiredoxin active site (prepared using LigPlot+ [60]).

**Figure 10 ijms-24-01410-f010:**
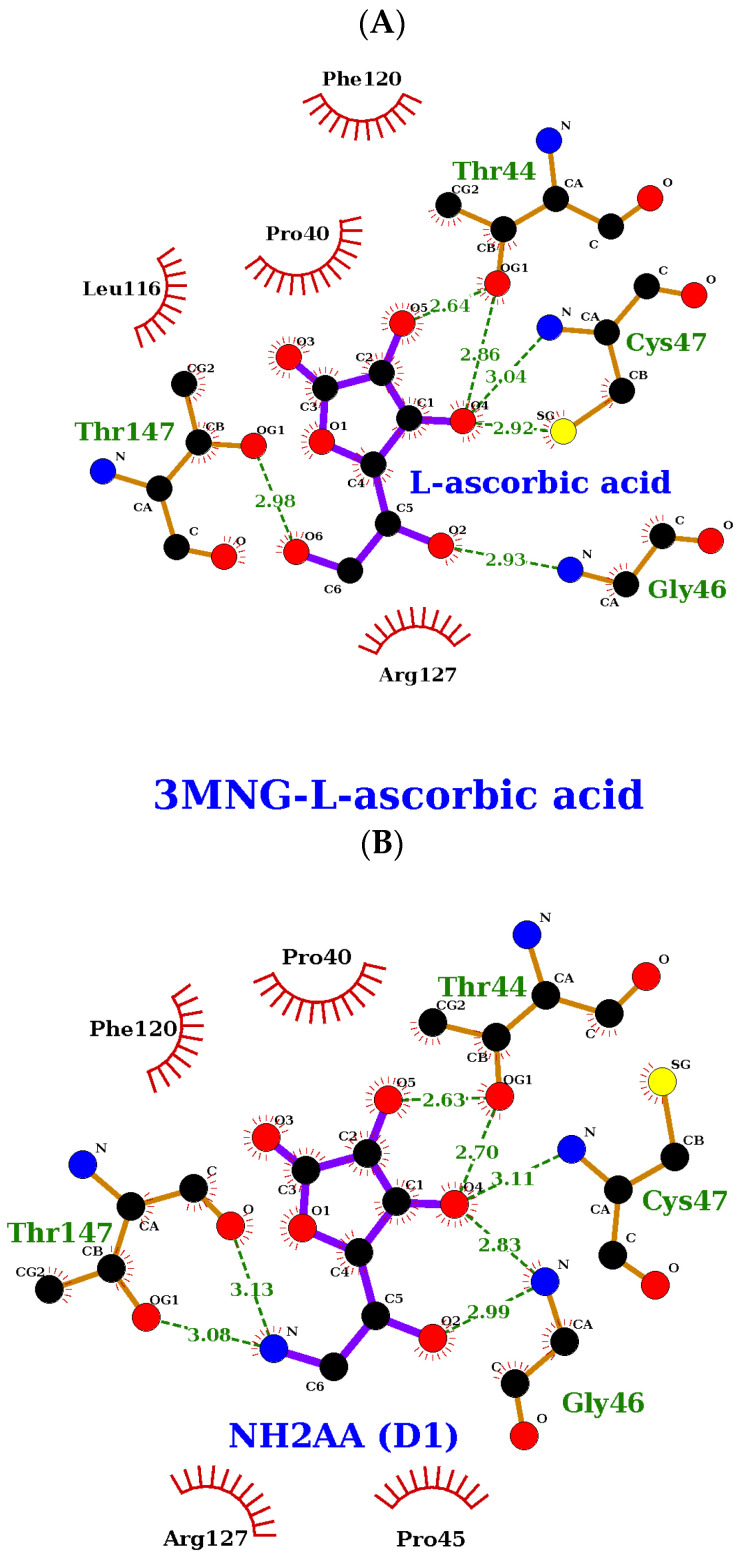
AA (**A**), NH_2_AA (D1) (**B**), and CH_3_NHAA (D2) (**C**) docked to 3MNG.

**Figure 11 ijms-24-01410-f011:**
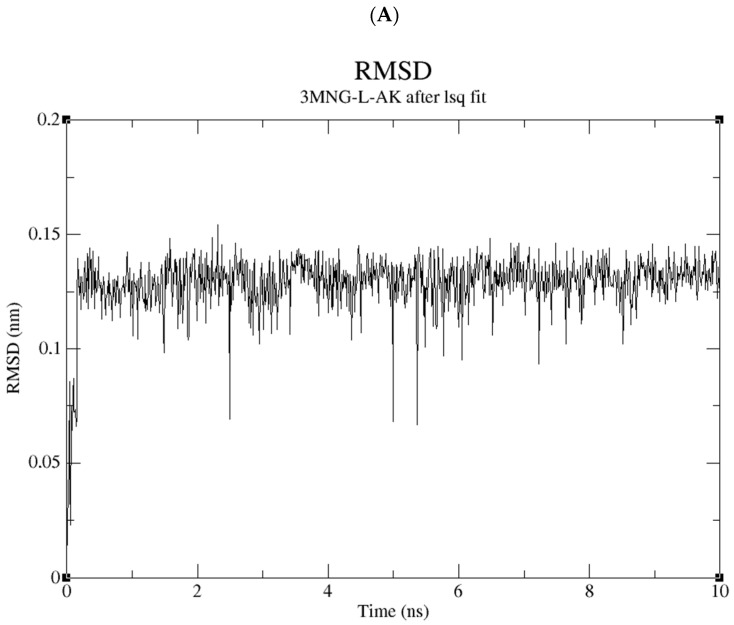
RMSD of AA (**A**), NH_2_AA (D1) (**B**), and CH_3_NHAA (D2) (**C**).

**Table 1 ijms-24-01410-t001:** The docking scores and structural data for dithiothreitol (DTT), AA, NH_2_AA (D1), and CH_3_NHAA (D2). The table lists the number of aromatic rings (aromatic rings), the number of hydrogen bond acceptors (Hb acceptors), the number of hydrogen bond donors (Hb donors), the number of rotatable bonds (rotatable bonds), the molecular weight (MW), the number of bonds (bonds), and the number of atoms (atoms) for each analyzed constituent.

Ligand	Best Score	Aromatic Rings	Hb	Hb	Rotatable Bonds	MW	Bonds	Atoms
			Acceptors	Donors				
DTT	−26.335	0	2	2	0	152.228	10	10
AA	−26.641	0	6	4	2	176.130	16	16
D1	−25.576	0	5	6	2	176.153	18	18
D2	−26.521	0	5	5	3	190.180	18	18

**Table 2 ijms-24-01410-t002:** MRM parameters for MS/MS analysis. The table lists the recording mode (mode), *m*/*z* precursor ion, fragmentor (frag), *m*/*z* fragment ions and collision energies (CE) for AA, NH_2_AA (D1), and CH_3_NHAA (D2).

Constituent	Mode	*m*/*z* Precursor Ion	frag/V	*m*/*z* Fragment Ions	CE/V
AA	-	174.8	90	130.3	6
				114.5	6
				86.4	18
D1	+	175.9	80	157.8	4
				140.8	10
				56.1	14
D2	+	189.9	110	171.8	6
				140.7	10
				70.1	16

Quantifier ions are underlined.

## Data Availability

The data presented in this study are available in the article or supplementary material. Additional data are available on request from the corresponding author.

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
