# Peer review of "Antioxidant Activity, Stability in Aqueous Medium and Molecular Docking/Dynamics Study of 6-Amino- and N-Methyl-6-amino-L-ascorbic Acid"

_ijms, 2023, doi:10.3390/ijms24021410_

Round 1
Reviewer 1 Report
The Manuscript by L.S. Martinovic, N. Birkic, V. Miletic, R. Antolovic, D. Stanfel, K. Wittine “Chemical stability of L-ascorbic acid is enhanced by bioisosteric replacement of OH to NH2” describes the study of antioxidant activity and chemical stability of 6-amino-6-deoxyascorbic acid and 6-methylamino-6-deoxyascorbic acid. The search for effective and readily accessible antioxidants possessing high stability on storage represents an urgent task. The possibility to obtain novel ascorbic acid-based antioxidants that retain biological targeting and possess improved properties was demonstrated convincingly. The advantages of the work include an extensive study of antioxidant properties and stability of amino-analogs of ascorbic acid and the use of molecular docking and molecular dynamics simulation to demonstrate the bioisosterism of ascorbic acid and corresponding 6-amino-6-deoxy derivatives.
However, the manuscript lacks of scientific novelty. Structural analogs of ascorbic acid discussed in manuscript had been described in 1980-1990s. Authors postulated that “extended research on NH-bioisosters of L-ascorbic acid is necessary in order to further evaluate their cosmetic potential…”. Thus, it is necessary to perform at least preliminary assay of studied compounds as components of cosmetic compositions. Moreover, it will be fruitful to synthesize several new simple-in-structure ascorbic acid bioisosters (e.g., containing the residues of ethylenediamine, hydroxylamine and so on…) and assess their stability and antioxidant properties.
The reference [26] on Croatica Chemica Acta 1989, 62, 537-544 should be corrected.
I believe that the synthesis of some new ascorbic acid analogs as well as testing of ascorbic acid bioisosters as compositions ingredients, incl. in vivo experiments, will strongly improve the manuscript and make it more interesting for a wide auditory of International Journal of Molecular Sciences.
Author Response
The Manuscript by L.S. Martinovic, N. Birkic, V. Miletic, R. Antolovic, D. Stanfel, K. Wittine “Chemical stability of L-ascorbic acid is enhanced by bioisosteric replacement of OH to NH2” describes the study of antioxidant activity and chemical stability of 6-amino-6-deoxyascorbic acid and 6-methylamino-6-deoxyascorbic acid. The search for effective and readily accessible antioxidants possessing high stability on storage represents an urgent task. The possibility to obtain novel ascorbic acid-based antioxidants that retain biological targeting and possess improved properties was demonstrated convincingly. The advantages of the work include an extensive study of antioxidant properties and stability of amino-analogs of ascorbic acid and the use of molecular docking and molecular dynamics simulation to demonstrate the bioisosterism of ascorbic acid and corresponding 6-amino-6-deoxy derivatives.
However, the manuscript lacks of scientific novelty. Structural analogs of ascorbic acid discussed in manuscript had been described in 1980-1990s. Authors postulated that “extended research on NH-bioisosters of L-ascorbic acid is necessary in order to further evaluate their cosmetic potential…”. Thus, it is necessary to perform at least preliminary assay of studied compounds as components of cosmetic compositions. Moreover, it will be fruitful to synthesize several new simple-in-structure ascorbic acid bioisosters (e.g., containing the residues of ethylenediamine, hydroxylamine and so on…) and assess their stability and antioxidant properties.
The reference [26] on Croatica Chemica Acta 1989, 62, 537-544 should be corrected.
Reference was corrected.
I believe that the synthesis of some new ascorbic acid analogs as well as testing of ascorbic acid bioisosters as compositions ingredients, incl. in vivo experiments, will strongly improve the manuscript and make it more interesting for a wide auditory of International Journal of Molecular Sciences.
The authors agree with the comment of reviewer that the structural analogues of ascorbic acid discussed in the manuscript had been described in 1980-1990. There are many topics in the literature that were forgotten or not enough attention was given to them, especially from a biological point of view. Still, we cannot agree that the manuscript does not contribute scientifically enough with the presented results and needs more scientific novelty. The manuscript has clearly shown the structural homology of the ascorbic acid and two analogues and their improved stability under described acidic conditions; moreover, the molecular docking/dynamics study suggests binding to wild-type of the human peroxiredoxin which is important for further evaluation of these and similar compounds on intracellular antioxidant systems. We agree that further derivatization is needed, and the new in vitro and in vivo experiments to define the mode of action. These and other steps in further characterization are foreseen for future work. Authors believe that the manuscript in this form will awaken the readers' interest in the broad specific scientific auditorium. Scientific work is always unfinished and endless in terms of possibilities but this is the same force that drives us to keep going.
We apologize for using expression „new, newly“. Synthesized derivatives per se are not novel. We removed the expression accordingly.
Author Response
I would like to appreciate the research work done by this group. The research work is up to the mark but the results are reported without providing any scientific reason or factors. The paper need some major changes.
General comments:
- The tittle of the paper did not comprehensively represent the type of research work reported as no chemical stability or the types of chemical interactions are discussed in the paper.
The title was slightly changed: “Stability of ascorbic acid in aqueous media is enhanced by bioisosteric replacement of OH to NH2”. We didn’t put molecular docking/dynamics in the title – this we consider an added value.
- The sentences are too long, which should be precise and informative.
The sentences were shortened where convenient.
- Abbreviations must be explained on first use and full name should not be used again in the text.
This was corrected.
- Most of the references are old; up to date reference should be added to support the results.
Introduction part was expanded to explain the biologic implications of improved antioxidant stability. This way, the referencing was updated and doubled. Moreover, explanation of biologic implications of improved antioxidant stability guides to reader to significance of performed research and future experimental possibilities.
Specific comments:
1.The PH value of healthy skin is in the range of 4.0 to 6.0 as mentioned in line 280-281 page 13. Then why the researchers have selected the PH value of 2.8 for acidic medium in their studies?
We used pH 2.8 in acidic medium in order to analyze both derivatives in the same form. pH 2.8 is below pKa1 value of both derivatives. As mentioned in the manuscript, the pKa1 value of derivative D1 is 3.7, and the pKa1 value of derivative D2 is 3.9 (calculations made with Percepta 2019.1.0 program (ACD Labs), and at pH values lower than their pKa1, the molecules are positively charged. Positively charged molecules in topical formulations may lead to improved penetration due to increased interaction with the negatively charged membrane.
- Explain the reason for high IC50 value of D1 and D2 then reported MAP and SAP derivatives by ABTS method.
D1 and D2 showed low IC50 values comparable to the IC50 values of ascorbic acid. Furthermore, the high values of SAP and MAP derivatives are the result of their stability. Namely, Liu et. al. 2020 (reference cited in manuscript) showed that the antioxidant activity of AA derivatives is inversely proportional to the degree of their stability. In the aforementioned work, other forms of AA derivatives were examined, and the final answer could be obtained by conducting the same research on NH derivatives.
https://pubs.acs.org/doi/10.1021/acsomega.0c04318
- What’s the reason for prolong stability of D2 until 15th day of incubation in ABTS method and less stability of DI
Stability could be influenced by electron-donating methyl group. Stability is also pH dependent. Careful track of degradation products should be performed in order to make a valid conclusion. Of importance is of course the mechanism by which the antioxidant function (both are ascorbic acid derivatives) – nevertheless, everything should be carefully experimentally checked before making a conclusion. On the other hand, ABTS method is generally rapid and can be used over a wide range of pH values.
- Why the stability of D1 and D2 at +20C in acidic medium and +4C in neutral medium was not studies for better comparison in MS/MS studies.
Scientific research has shown that DPPH and ABTS radicals are pH-dependent and have different areas of optimal pH action that do not include the low pH used in this research (pH below the pKa values of D1 and D2 derivatives and therefore these analyses were not performed. DPPH and ABTS antioxidant activity stability study was used as a preliminary point for further MS/MS stability study. More accurate analyzes were obtained by MS/MS analysis.
This explanation has been added to the discussion.
see. https://doi.org/10.1016/j.jfca.2013.02.004
- The reason for more stability of D1 then D2 in neutral medium at +20c must be explained.
Stability in neutral medium could be explained by hydrogen bond network stabilization. D1 has NH2 while D2 has CH3NH (6 Hb donors for D1 vs. 5 Hb donors for D2). But this is an assumption,
- As Reported in this paper: the stability of D2 is higher as measured by the spectrophotometric methods but D1 is more stable by MS/MS studies. Provide explaination.
This observation is correct and it is precisely for this reason that the MSMS analysis was performed. The ABTS radical is an unstable radical that enables rapid analysis of antioxidant activity (and if observed over a longer period; antioxidant stability). Precisely because of the instability of the radicals, the results may deviate from the actual values. The results of the spectrophotometric methods selected in this paper served as starting points for further more accurate stability analysis made using the MSMS method.
Reviewer 3 Report
Well written manuscript with clear and consistent results. I would recommend two slight changes to the manuscript.
-Consider expanding the introduction after the sentence on line 68 to explain the biologic applications of improved antioxidant stability. The discussion at the end does a good job of this (Section 3.1) and a brief inclusion in the introduction to explain why this research is of "considerable importance" will orient the author to understand the context better.
-Please include the IC50 values for SAP and MAP in the text on line 97. This will highlight the superiority of D1 and D2 to SAP/MAP without requiring the reader to download the supplemental material.
Author Response
Reviewer 3
Well written manuscript with clear and consistent results. I would recommend two slight changes to the manuscript.
-Consider expanding the introduction after the sentence on line 68 to explain the biologic applications of improved antioxidant stability. The discussion at the end does a good job of this (Section 3.1) and a brief inclusion in the introduction to explain why this research is of "considerable importance" will orient the author to understand the context better.
We thank the reviewer for the comment and we have introduced required changes to the manuscript. Introduction part was expanded to explain the biologic implications of improved antioxidant stability which also guides to reader to future experimental possibilities.
-Please include the IC50 values for SAP and MAP in the text on line 97. This will highlight the superiority of D1 and D2 to SAP/MAP without requiring the reader to download the supplemental material.
This was added to the manuscript.
Round 2
Reviewer 1 Report
In revised version of manuscript, derivatives of ascorbic acid (D1 and D2) obtained and characterized previously are not named as novel compounds. Hovewer urgency and scientific soundness of the manuscript can still raise some doubts. But I do agree with authors' notation that the paper can awaken the interest of readers to the useful properties of ascorbic acid derivatives, to methods for analysis of these properties. Present study can serve as a starting point for further research. Now I feel that the manuscript can be published in International Journal of Molecular Sciences. I hope that in the future authors will provide some new results arising from the present work.
Author Response
We thank the reviewer for the comments and suggestions. In the future work we will expand the library of C6 substituted L-ascorbic acid derivatives to asses intracellular antioxidant and anti-inflammatory potential. As suggested by reviewer, we will also try to prepare hydroxylamine and ethylendiamine derivative.
Reviewer 2 Report
The tittle of the paper is does not fully explain the type of research work. The present tittle looks looks like a statement rather then a tittle.
All the answers of the previously provided comments should be the part of the research paper. As it clears many questions regarding the research work.
Author Response
The tittle of the paper is does not fully explain the type of research work. The present tittle looks looks like a statement rather then a tittle.
The title of the manuscript was changed to: Antioxidant activity, stability in aqueous medium and molecular docking/dynamics study of 6-amino- and N-methyl-6-amino-L-ascorbic acid. We hope that the new title reflects the work covered by the manuscript.
All the answers of the previously provided comments should be the part of the research paper. As it clears many questions regarding the research work.
We would like to thank the reviewer for his kind suggestions for improvements. We have incorporated answers to comments to the manuscript.